# Review of the Anti-Pollution Performance of Triple-Layer GM/GCL/AL Composite Liners

**DOI:** 10.3390/membranes12100922

**Published:** 2022-09-23

**Authors:** Jia Li, Chuhao Huang, Jingwei Zhang, Zhanguang Zhang

**Affiliations:** School of Water Conservancy and Civil Engineering, Zhengzhou University, Zhengzhou 450002, China

**Keywords:** GM/GCL/AL, triple-layer composite liner, anti-pollution performance, transport model, breakthrough time

## Abstract

Landfill leachates contain several types of pollutants and complex components, which pollute soils and groundwater. To compensate for the limitations of single-layer and double-layer liners, a triple-layer liner system composed of a geomembrane (GM), geosynthetic clay liner (GCL), and attenuation layer (AL) was invented and widely used in landfill anti-pollution systems. In this paper, the available literature on triple-layer GM/GCL/AL composite liners is summarized. First, the four main transport processes of pollutants through the composite liner, including convection, diffusion, adsorption, and degradation, were analyzed, and the anti-pollution performances were evaluated. According to this, the pollutant transport model considering the transport activity and transport state was classified, and the solution methods were summarized. Finally, the breakthrough time expressions of the composite liners were determined, which provided a base for evaluating their long-term performance and predicting the service life. The purpose of this literature review is to scientifically evaluate the anti-pollution performance of GM/GCL/AL and provide a scientific base and theoretical guidance for extending its application.

## 1. Introduction

Soil and groundwater pollution is a global problem, with more than five million contaminated sites around the world [1], and the municipal solid waste (MSW) landfills constitute a potential major threat. A wide range of toxic inorganic and volatile organic compounds (VOCs) are contained in MSW leachate, which pose an environmental threat to groundwater sources. Accordingly, the liner system is one of the most important elements of the modern engineered landfill.

The liner system is made of impermeable materials, such as geomembrane (GM), geosynthetic clay liner (GCL), compacted clay liner (CCL), and attenuation layer (AL). Among them, GM is one of the most widely used geosynthetic materials because of its excellent anti-seepage performance. GM can effectively inhibit the leakage of inorganic pollutants, but is not efficient in blocking the transport of organic pollutants [2]. In order to overcome the defects, some of the liners are combined with GM to form composite liners. In the past, CCL was often used as the GM underlying liner, but due to the high cost and inconvenient construction of CCL, GCL has more recently emerged as a substitute. However, some studies have shown that GM/GCL has a mediocre barrier effect to certain pollutants, but this can be improved by attaching AL, so GM/GCL/AL is proving to be a viable liner system (Figure 1), and the study of its anti-pollution performance is of great theoretical and practical significance to further applications.

This paper reviews the relevant studies and analyzes the research trends and progress in recent decades. As can be seen from Figure 2a–d, the number of studies and researchers showed an overall upward trend, and with a peak in the past 5 years. During this period, scholars have proposed many examined approaches. Moreover, the research of China, Canada, and the United States accounts for the majority of the current research, which is due to the large number of landfills in these countries, and the related policy norms are also more complete. In terms of anti-pollution liner types, most of the research has focused on GM single-layer and double-layer liners, while research on triple-layer composite liners, especially GM/GCL/AL, is relatively lacking.

Therefore, the anti-pollution performance of GM/GCL/AL composite liners was comprehensively evaluated in this paper, and the research process was presented in the form of a flow chart (Figure 3). In general, the anti-pollution performance of liners was evaluated in three aspects, which are pollutants’ transport mechanism, transport model, and breakthrough time. The research methods and important parameters are also summarized in Table 1. Through the evaluation of the above indicators, some suggestions and perspectives were put forward, which provides a basis for the application of the liners.

## 2. Convection of Pollutants through GM/GCL/AL

The convection of pollutants through the composite liner is mainly through leakage through GM defects, which can be assessed by the leakage rate. Although a small amount of pollutants are transported through GCL/Al, the two layers are still effective in inhibiting the convection of pollutants through the GM.

### 2.1. GM Defects

Because pollutants leak mainly through the GM, its integrity is the decisive factor in controlling the leakage. However, in an actual landfill, GM defects cannot be avoided because of transportation or construction inadequacies and wrinkles formed during GM installation. Rowe [2] proposed that the leakage rate of pollutants in the defective GM is a sum of: (1) the leakage rate through the holes on the flat GM and (2) the leakage rate through the holes connected to the wrinkles on the GM.

The formula to calculate the leakage rate has been derived in several studies [3,4,5,21,22], most of which were based on experience and experimental data, whereas few were derived theoretically. Giroud [3] classified the formulae of the leakage rate calculation according to the interface condition, log(Ki/Ks), where Ki and Ks represent the permeability coefficients of the interface and underlying soil cushion or GCL, respectively. Table 2 lists the applicable conditions and calculation methods for the various leakage rate formula.

The method available in this paper allows the calculation of leakage rates for conditions beyond those for which experimental verification exists. The reader is therefore urged to use the equations carefully and to apply judgement in interpreting calculation results. At the same time, researchers are encouraged to conduct experiments to quantify leakage rates through composite liners and thereby contribute to the body of available data.

However, it is worth noting that the above formulae can only be applied to a flat GM. When the GM is exposed to solar radiation during installation, it bends upward to form wrinkles [23,24,25,26,27,28]. Because of the large number of wrinkles on GM (the coverage rate of wrinkles can reach 30% or even higher) and most of them being hydraulically connected, the defects most likely combine with them [29,30,31,32,33,34]. To calculate the leakage rate of the hole combined with the wrinkle (Figure 4), the length of the wrinkle is assumed to be significantly greater than its width, and that the leachate first moves laterally in the transfer layer, beyond the boundary of the wrinkle to a wetting radius, after which it moves longitudinally in the underlying soil layer. Rowe et al. [21] provided a formula to calculate leakage rate as:$$Q_{c}=2L\left[\left(kb+{\left(kD\theta \right)}^{0.5}\right)\right]h_{d}/D$$
where *L* (m) is the length of the wrinkle; *2b* (m) is the width of the wrinkle; *k* (m^2^/s) is the permeability of the clay liner; *D* (m) is the thickness of the clay liner; *θ* (m^2^/s) is the interfacial hydraulic conductivity between the GM layer and clay liner; *h_d_* (m) is the water head loss in the composite liner.

Rowe et al. [35] calculated the leakage rate of the leachate in a composite liner made of GM and clay, and showed that the effect of wrinkles is significant, and the length of hydraulically connected wrinkles significantly affects the leakage rate. Zhang [36] further studied the influence of the length of hydraulically connected wrinkles on the leakage rate of composite liners, and showed that the leakage rate of the hole with the wrinkle was 10–5000 times more than that of the hole on the flat GM. Therefore, wrinkles are important because of the increased potential for contaminant migration through a hole in the GM at or near the wrinkle. There is also increased potential for the development of future holes due to stress cracking at points of high tensile stress in the wrinkle. While it is not essential to eliminate all of the wrinkles, the presence of wrinkles should be considered in calculating leakage through composite liners in the design. The construction procedure should be adopted to limit the length of connected wrinkles present at the time the GM is covered.

### 2.2. Interface Transmissivity

Although the leakage rate is mainly controlled by GM defects, the interface transmissivity between the GM and the underlying medium also influences the leakage rate of composite liners. Chai et al. [37] described two methods for calculating the value of transmissivity: one is by inverse calculation from the results of flow test experiments and the other is by directly measuring the GM/underlying layer interface transmissivity. For example, Barroso [38] measured the leakage rate of a GM (with holes)/GCL/soil-based composite liner through an indoor leakage test. The interface transmissivity was obtained using the formula proposed by Rowe and Jabin [39] for the leakage rate of leachate in the wetted radius of the composite liner. The obtained value of the GM/GCL was 3.2 × 10^−11^ to 5 × 10^−11^ m^2^/s.

Touze-Foltz et al. [40] directly measured the interface transmissivity of a GM with holes and a GCL using an indoor test. The interface transmissivity formula for the GM/GCL was deduced as follows:$$\theta =\frac{aln\left(R_{2}/r_{0}\right)ln\left(h_{0}/h_{1}\right)}{2\pi \Delta t}$$
where *r*_0_ (m) is the radius of the circular hole; *R*_2_ (m) is the outer diameter of the sample; *a* (m^2^) is the cross-sectional area of the burette; *h*_0_ (m) is the initial water-head difference; *h*_1_ (m) is the final water-head difference; and Δ*t* (s) is the time interval. The obtained interface transmissivity was 6 × 10^−12^ to 2 × 10^−10^ m^2^/s.

Chai et al. [41] conducted large-scale and long-term leakage tests on the leachate using a composite liner made of defective GM and clay layers. They showed that when the defect was a circular hole of 5 mm radius, the range of interface transmissivity under different loads was 1.3 × 10^−6^ to 1 × 10^−7^ m^2^/s; when the defect was a 1 × 30 mm rectangular hole, the range became 1.3 × 10^−10^ to 1.4 × 10^−8^ m^2^/s. Considering the influence of load *p*′ on the leakage rate Q of the GM with holes, the empirical formula of the effect of *p*′ on *θ* is given as follows:$$\theta =\theta_{0}\frac{1}{{\left(1+p^{\prime }/p_{a}\right)}^{2.5}}$$
where *θ*_0_ (m^2^/s) is the interface transmissivity when *p*′ = 0 and *p_a_* is the atmospheric pressure.

Additionally, the conditions of contact between the GM and underlying layer play a decisive role in the interface transmissivity. Imperfect contact allows for a transmissive zone between the two liners, and hence, once fluid has migrated through the defect, it can then move laterally in the interface between the two liners before fully percolating through the clay liner. Giroud [3] defined the good and poor contact conditions between the GM and the underlying soil layer; where good contact indicates that the GM is almost wrinkle-free, the underlying soil layer is fully compact, and the surface is flat, while poor contact indicates that the GM has a certain number of wrinkles, and the underlying soil layer is uneven and not fully compact.

Rowe [42] used the empirical formula derived by Giroud [3] to obtain the empirical relationship between the interface transmissivity of the composite liner and permeability coefficient of the underlying GM liner:

① Good contact: $\log_{10}\theta =0.07+1.036\left(\log_{10}k_{L}\right)+0.0180{\left(\log_{10}k_{L}\right)}^{2}$

② Poor contact: $\log_{10}\theta =1.15+1.092\left(\log_{10}k_{L}\right)+0.0207{\left(\log_{10}k_{L}\right)}^{2}$ where *K_L_* (m^2^/s) is the permeability coefficient of the underlying layer. When *K_L_* is 1 × 10^−9^ m^2^/s for a typical CCL, the corresponding interface transmissivity (*θ*) is 1.6 × 10^−8^ m^2^/s for a good contact and 1 × 10^−7^ m^2^/s for a poor contact. It was observed that the interface transmissivity of a good contact is lower by nearly one order of magnitude than that of a poor contact.

### 2.3. GCL/AL Consolidation Characteristics

As mentioned above, pollutants mainly leak through the defective GM of a composite liner, and the defect type, liner contact conditions, external load, and properties of the underlying layer would affect the leakage rate. Therefore, for the triple-layer GM/GCL/AL composite liner reviewed in this paper, the characteristics of GM, GCL, and AL need to be considered when calculating the leakage rate, in which the consolidation characteristics are the most important factors.

AL is a natural soil layer. When consolidated, water flows out of the soil, and the solid particles and fluids move vertically through the upper and lower surfaces of the unit. Following the mass conservation principle for the transport of pollutants in soil, Peter and Smith [43] obtained the mass conservation equation of the liquid and solid phases through the unit. The Darcy rate for the consolidated soil can be obtained by the following derivation:$$q=-\frac{k_{v}}{\rho_{f}g}\left(\frac{\partial u_{h}}{\partial z}+\frac{\partial p}{\partial z}\right)$$
where *k_v_* (m^2^/s) is the permeability coefficient of the soil; *ρ_f_* (kg/m^3^) is the density of the liquid phase; *μ* is the viscosity coefficient; *g* (m/s^2^) is the acceleration due to gravity; *u_h_* (kPa) is the total pressure of the liquid phase; and *p* (kPa) is the excess pore water pressure caused by consolidation. The first term on the right of the equation represents the Darcy velocity caused by the hydraulic gradient, and the second term represents the Darcy velocity caused by consolidation.

Based on this, Peters and Smith [43] proposed consolidation equations for small and large deformations using the Terzaghi effective stress theory. For small deformations, the stress change in the soil can be assumed as constant without considering the effect of velocity caused by the hydraulic gradient:$$\frac{1+e}{a_{v}}\frac{\partial \left[\frac{k_{v}}{\rho_{f}g}\frac{\partial p}{\partial z}\right]}{\partial z}+\frac{\partial \sigma }{\partial t}=\frac{\partial p}{\partial t}$$
where *e* is the porosity ratio; *a_v_* is compression coefficient; and *σ* (kPa) is the total vertical pressure.

In case of consolidation by large deformations, porosity varies with time and depth (porosity varies uniformly with depth for small deformations). Consolidation can be expressed by the equation for change in the porosity ratio:$$\frac{\partial e}{\partial t}=\pm \left(1+e_{0}\right)\left(\frac{\rho_{s}}{\rho_{f}}-1\right)\frac{d}{de}\left(\frac{k_{v}}{1+e}\right)\frac{\partial e}{\partial z}+\left(1+e_{0}\right)\frac{\partial }{\partial z}\left(\frac{k_{v}}{\rho_{f}ga_{v}}\frac{1+e_{0}}{1+e}\frac{\partial e}{\partial z}\right)$$
where *e_0_* is the initial porosity ratio and *ρ_s_* (kg/m^3^) is the density of the solid phase.

Many scholars have studied the consolidation characteristics of GCL. Kang and Shackelford [44] tested GCL repeat samples in a flexible-walled instrument and measured the relevant parameters. They showed that the consolidation coefficient of the GCL ranged from 5.2 × 10^−10^ m^2^/s to 2.1 × 10^−9^ m^2^/s, and the hydraulic conductivity coefficient k was ≤5.0 × 10^−9^ cm/s, indicating that the consolidation behavior of the GCL is consistent with that in small deformations. The influence of consolidation characteristics of the GCL on the semi-permeable membrane behavior has been further studied [45]. Such behavior reflects the relative barrier effect of liquid solutes, which is usually quantified by membrane efficiency (ω), where a larger ω indicates a smaller liquid flow through the GCL. The experimental results showed that the membrane behavior of the GCL was significantly enhanced by consolidation. Lu et al. [46] evaluated the effect of pore water salinity on the consolidation characteristics of the GCL using one-dimensional consolidation testing equipment, and showed that the compression and expansion indexes decreased with increasing liquid salinity, while the characteristics of the elastic hysteretic volume change of the GCL were observed in the permeable consolidation test results.

Thus, it is inferred that the convective transport of pollutants in the composite liner is mainly influenced by leakage through the GM defects, and can be evaluated by estimating the leakage rate. The interface transmissivity and consolidation characteristics are also important factors affecting the leakage rate. The anti-pollution performance of the liner can be preliminarily evaluated by the leakage rate. It is suggested to consider the above factors and select an appropriate formula to calculate the leakage rate for evaluation.

Besides, the study of the convection characteristics of pollutants through the composite liner is lacking in some aspects: (1) The leakage rate of the GM with different defects was calculated based on a single defect, and it is necessary to consider the combined effect of various defects on the leakage rate. (2) Presently, research on interface transmissivity is scarce Most studies are being conducted on the GM double-layer composite liner, with little on the GM/GCL/AL liner. (3) The influence of the consolidation characteristics of GCL/AL consolidation on the leakage rate of the composite liner has not been systematically studied.

## 3. Diffusion of Pollutants through GM/GCL/AL

The transport of pollutants, such as volatile organic compounds (VOC), is dominated by diffusion, which can be evaluated by the diffusion coefficient. The diffusion mechanisms of pollutants differ in different media.

### 3.1. Diffusion of Pollutants through GM

The diffusion of pollutants in the GM occurs through the pore space between polymers [47]. This process was simulated using Fick’s law:$$J_{G}=-D_{g}\frac{\partial C_{g}}{\partial z}$$
where *J_G_* [kg/(m^2^·s)] is the mass flux of pollutants through the GM and *D_g_* (m^2^/s) is the diffusion coefficient of pollutants in the GM. The transient diffusion equation can be obtained by applying the law of mass conservation:$$\frac{\partial C_{g}\left(z,t\right)}{\partial t}=D_{g}\frac{\partial^{2}C_{g}\left(z,t\right)}{\partial z^{2}}$$

The diffusion of organic pollutants through the GM has been studied to reveal that the diffusion coefficients of polar molecules, such as water molecules and inorganic ions in the GM, are very small, while those of the organic pollutants are quite large [47]. Eun et al. [48] and Mcwatters [6] showed that the diffusion rate of organic compounds in the GM is quite high. Mcwatters et al. [7] further showed that the breakthrough time of organic chemical transport through a GM was only 24 h, and the pollutant flux reached a steady state after approximately one week. In contrast, Saheli et al. [8] showed that inorganic pollutants cannot diffuse through the GM. Therefore, the diffusion of organic pollutants through GM is the focus of most studies.

The diffusion coefficients of organic pollutants in the GM mainly depend on the pollutant properties. The affinity between the pollutants and GM is often expressed by the n-octanol/water partition coefficient (*log K_ow_*); *log K_ow_* refers to the ratio of the concentration of an organic pollutant in the n-octanol and water phases. The range of *log K_ow_* is generally between −3 and 7, and compounds with lower *log K_ow_* values are more hydrophilic. For organic pollutants, Dg increases with *log K_ow_*. Rowe et al. [49] obtained a simple empirical formula for the geometric diffusion coefficient based on previous experimental data:$$\log D_{g}=-12.3624+0.9205\;logK_{\mathrm{ow}\;}-0.3424{\left(\log K_{\mathrm{ow}}\right)}^{2}$$

Through the above formula, the approximate diffusion coefficient of organic pollutants can be calculated, and then the diffusion characteristics can be evaluated. However, in order to measure the diffusion coefficient more accurately, it is still recommended to measure it through indoor tests. The technique that best matches actual situations uses a two-compartment diffusion cell, where contaminant diffusion is monitored from the source to the receptor. When equilibrium is reached, the value of S_gf_ can be deduced from mass balance considerations. Scholars have tested the diffusion coefficients of some organic pollutants and discussed their influencing factors.

Rowe et al. [49] experimentally studied the influence of surface fluorination on GM performance. Through adsorption and diffusion tests of VOCs on traditional 1.5 mm thick high density polyethylene (HDPE) GM and surface-fluorinated GM (f-HDPE), it was found that the diffusion coefficient and permeability of f-HDPE were 1.5–4.5 times lower than those of the traditional HDPE GM. Rowe et al. [50] studied the effect of surface fluorination on the diffusion characteristics of benzene, toluene, ethylbenzene, xylene, and other organic substances (BTEX). It was observed that at 22 °C and 6 °C, the permeability of the BTEX solution in the f-HDPE GM is 2.4 times and 1.8 times lower than that in the ordinary HDPE GM, respectively.

Therefore, the characteristics of the pollutants are the main factors affecting the diffusion characteristics; additionally, the GM properties and ambient temperature are influential factors, and the influence is significant under specific conditions. It is suggested that the above factors should be taken into account as far as possible in practical application.

### 3.2. Diffusion of Pollutants through GCL

The diffusion of pollutants in the GCL is described by Fick’s law:$$J_{GCL}=-n_{e}D^{*}\frac{\partial C}{\partial z}=-D_{p}\frac{\partial C}{\partial z}$$
where *J_GCL_* is the flux of pollutants through the GCL and *D_P_* = *n_e_D** is the diffusion coefficient of porous media. Lake and Rowe [51] found that *D_P_* is a linear function of the final porosity ratio *e_B_* of the GCL for Na^+^ and Cl^−^:$$D_{p}\approx \left(1.02e_{B}-0.89\right)\times 10^{-10}$$

The diffusion coefficient linearly increases with the increasing pore ratio because of the increase in the free pore space. Moreover, the concentration of the pollutant solution, vertical pressure exerted by GCL hydration, and ambient temperature can all affect the diffusion coefficient of pollutants in the GCL. For example, Lake and Rowe [51] showed that when the concentration of NaCl solution increases from 4.6 g/L to 114 g/L, the diffusion coefficient of Cl^−^ through the GCL increased from 1 × 10^−10^ m^2^/s to 3 × 10^−10^ m^2^/s. Rowe et al. [52] showed that when the temperature increased from 7 to 22 °C, the effective diffusion coefficient of benzene in GCL increased from 2.2 × 10^−10^ to (3.7–4.0) × 10^−10^ m^2^/s.

Previous studies suggest that the diffusion coefficient of pollutants in the GCL is similar to that in soil, so GCL and AL can be considered as a whole in the diffusion effect. In addition, the diffusion coefficient of organic pollutants ((3.2–5.6) × 10^−10^ m^2^/s) is greater than that of inorganic pollutants ((1.5–3.0) × 10^−10^ m^2^/s).

### 3.3. Diffusion of Pollutants through AL

The pollutants diffuse by molecular diffusion in AL [53], which is caused by their irregular thermal movement (Brownian motion), and they migrate from the high-concentration area to the low-concentration area.

Steady state diffusion in free solution is usually described by Fick’s first law:$$J_{D}=-D_{0}\frac{\partial C}{\partial z}$$
where *J_D_* [kg/(m^2^·s)] is the diffusion flux and *D*_0_ (m^2^/s) is the diffusion coefficient in the free solution.

The transient molecular diffusion problem is described by Fick’s second law:$$\frac{\partial C}{\partial t}=\frac{\partial }{\partial z}\left(D_{0}\frac{\partial C}{\partial z}\right)$$

Compared to diffusion in a free solution, the diffusion of pollutants through AL is more complex and relatively slow. The presence of solids reduces the cross-sectional area of the water flow as the solute diffuses only in the soil pores. However, because of the tortuosity of the surface of soil particles, the seepage path of solutes is more tortuous because of the surface tortuosity of the soil particles, which increases the diffusion distance of the pollutants. To accurately characterize the molecular diffusion mechanism of pollutants in AL, Shackelford [54] proposed a practical definition of the effective diffusion coefficient:$$D^{*}=\tau D_{0}$$
where *D** (m^2^/s) is the effective diffusion coefficient and *τ* is the tortuosity factor, which reflects the influence of soil particles, skeleton, and pore tortuosity on pollutant diffusion.

The apparent tortuosity factor of soil is usually 0.01–0.07 [55], and varies with soil types. For example, the tortuosity factor of montmorillonite is the smallest (0.15–0.20), followed by those of kaolin and clay that are similar (0.24–0.50 and 0.28–0.46, respectively), silty clay (0.08~0.30), sandy soil (0.28), and sand, which is the largest (0.47–0.51). The tortuosity factor of sand–bentonite has the largest variation range (0.01–0.84) [56,57,58,59,60].

The available methods to measure the effective diffusion coefficient of pollutants in soil include field test data inversion, vertical diffusion test, and horizontal diffusion test, which have been used to measure the effective diffusion coefficients of different pollutants through different types of soil. Xie [61] classified these values and demonstrated that the effective diffusion coefficient of inorganic pollutant ions through soil is in the range of (1–15) × 10^−10^ m^2^/s (average = (2–8) × 10^−10^ m^2^/s). The effective diffusion coefficient of organic compounds is generally up to 9 × 10^−10^ m^2^/s (average = (1.5–6) × 10^−10^ m^2^/s). Thus, it was inferred that the diffusion coefficient of organic pollutants in soil is smaller than that of inorganic ions.

In summary, the study of the diffusion of pollutants in a composite liner is mainly focused on organic pollutants, which can be evaluated by the diffusion coefficient. The method to obtain the diffusion coefficient is usually an indoor diffusion test, and the anti-pollution performance of the liner can be preliminarily evaluated by the diffusion coefficient. In addition, the chemical properties of the pollutants and the consolidation characteristics of the soil are also important factors, which should be considered in the actual situation.

However, these studies lack in the following aspects: (1) Studies on the diffusion characteristics of other pollutants are rare, and such studies are necessary to establish a database of the diffusion behavior of different pollutants. (2) Single-layer or double-layer composite liners are mainly studied, and diffusion coefficient of pollutants through the GM/GCL/AL triple-layer liner is rarely measured.

## 4. Adsorption of Pollutants through GM/GCL/AL

Adsorption is also a common mechanism in pollutant transport, which is mainly evaluated by adsorption coefficient. The adsorption of each layer is described below.

### 4.1. Adsorption of Pollutants through GM

Generally, contaminants are transported in non-defective GM by three continuous processes: ① distribution of pollutants between GM and leachate; ② diffusion of pollutants in GM; ③ distribution of pollutants between GM and the underlying media [62].

When the leachate and GM are in contact, the organic pollutants in the leachate are adsorbed by the GM, redistributing the organic pollutant concentration in the leachate on the surface between the leachate and the GM. This is essentially a process of adsorption of pollutants in the leachate by GM.

When the GM and leachate are in contact for a sufficient period of time, a definite relationship forms between the equilibrium concentration *C_g_* (g/L) of the leachate in GM and the equilibrium concentration *C_f_* (g/L) of the adjacent fluid. This relationship is usually expressed in terms of the Nernst distribution equation in a linear form (Henry’s law):$$C_{g}=S_{gf}C_{f}$$
where *S_gf_* is the partition coefficient, which is mainly affected by the chemical composition of the fluid, molecular structure of the GM, and the ambient temperature.

Henry’s law can also be used for the linear simulation of the distribution of pollutants between GM and the pore water of the underlying medium, i.e., the desorption of pollutants in the GM:$$C_{g}\prime =S_{gf}\prime C_{f}$$
where *S_gf_′* represents the distribution coefficient of pollutants in the GM and the pore fluid of the underlying medium. Currently, there are relatively few reports on the value of *S_gf_′*, and *S_gf_′* and *S_gf_* are generally considered equal [63].

The adsorption of pollutants in the GM is usually evaluated by the partition coefficient using different experimental techniques. Sangam and Rowe [64] used the immersion/adsorption method, and measured the partition coefficients of three chlorinated and four aromatic hydrocarbons through 2.0 mm thick HDPE GM using gas chromatography. Rowe et al. [65] studied the transport of six types of polychlorinated biphenyls homologues in HDPE GM by the immersion/adsorption test, and found that the partition coefficient ranged from 0.7 × 10^6^ to 7.5 × 10^6^. Zhang [36] also obtained the partition coefficient of bisphenol A (1.4) through 1.5 mm thick GM by the batch adsorption test. All of these obtained coefficients are valuable because they serve as a database for the evaluation of the adsorption performance of GM for different pollutants.

The similarity between the pollutants and GM mainly influences adsorption, and is expressed by *log K_ow_*, with minor influence from the molecular weight (Mw) of the pollutant. Rowe [63] summarized the empirical formula of the distribution coefficient based on the data from the previous tests (Table 3). It was shown that the partition coefficient of a VOC could be predicted using the available octanol/water partition coefficient (K_ow_). This provides a convenient means of estimating the partition coefficient of a VOC since the octanol/water partition coefficients of VOCs are readily available.

### 4.2. Adsorption of Pollutants through GCL

Pollutants are adsorbed through the GCL by the upper geotextile (capping layer), bentonite, and lower geotextile (load-bearing layer) layers [51]:$$K_{deq}=\frac{\left(m_{db}K_{db}+m_{dbGT1}K_{dGT1}+m_{dGT1}K_{dGT1}+m_{dGT2}K_{dGT2}\right)}{m_{db}+m_{dbGT1}+m_{dGT1}+m_{dGT2}}$$
where *m_db_*, *m_dbGT_*_1_, *m_dGT_*_1_, and *m_dGT_*_2_ (g) represent the masses of bentonite, geotextile contained in the bentonite soil layer per unit area, geotextile capping layer, and geotextile load-bearing layer, respectively. The corresponding *K_db_*, *K_dGT_*_1_, and *K_dGT_*_2_ represent the partition coefficients of the bentonite, capping, and load-bearing layers, respectively.

Pinto et al. [9] carried out batch adsorption tests of ammonium and bisphenol A solutions on bentonite and geotextiles, and showed that the adsorption capacity of bentonite for the two pollutants was significantly higher than that of geotextiles. Lake and Rowe [66] experimentally studied the adsorption characteristics of aromatic VOCs on each layer of a GCL. They showed that the adsorption capacity of the bentonite layer was almost negligible, and the overall adsorption capacity depended on the adsorption capacity of pollutants in the upper and lower geotextile layers. Saheli et al. [67] obtained the distribution coefficient of bisphenol A in GCL (10–16 mL/g). 

Additionally, Zhang et al. [68] performed a chemical analysis of landfill leachate before and after infiltration and showed that GCL has a strong adsorption capacity for cations and organic matter at the beginning of infiltration, which gradually reaches saturation and loses its adsorption capacity. Therefore, GCL is used as an anti-seepage barrier mainly for its low permeability rather than its adsorption of organic matter and harmful ions.

The above studies showed that adsorption of GCL is affected by several factors, including the properties of pollutants, bentonite, and geotextiles, and the ambient temperature. The relationship between the adsorption parameters of GCL on pollutants and the change in temperature is [69]:$$\frac{K_{dT1}}{K_{dT2}}=\theta^{\left(T_{1}-T_{2}\right)}$$
where *K_dT_*_1_ and *K_dT_*_2_ represent the distribution coefficients at temperatures, *T*_1_ and *T*_2_ (°C), respectively, and *θ* is determined by the following formula:$$\theta =\frac{X}{RT_{1}T_{2}}$$
where *X* (K) is the heat of the solution. *θ* is generally between 1.0 and 1.1. Rowe et al. [63] obtained *θ* values between 1.03 and 1.04 by fitting experimental data.

### 4.3. Adsorption of Pollutants through AL

When solid pollutants migrate to the soil pore space, they come in contact with soil particles and are captured by them through physical adsorption, ion exchange, and interface precipitation. The adsorption models can be divided according to various adsorption states into equilibrium adsorption, non-equilibrium adsorption, desorption, and competitive adsorption [10,11,70,71,72,73,74,75,76,77,78]. The classical adsorption models for different modes are summarized in Table 4.

The above adsorption models have been widely used in various studies. Among them, the linear adsorption model is the most convenient and often used in the derivation and solution of analytical solutions. However, the linear adsorption model assumes that there is no upper limit to adsorption, which is not consistent with the actual situation. Similarly, the Freundlich adsorption model also has this defect. The Langmuir model and the two-stage adsorption model are based on the upper limit of soil adsorption capacity, and compared to the Freundlich model, the expression of two-stage adsorption model is relatively easy, which can be used for the derivation of the analytical solution of the breakthrough time of pollutants. In addition, the competitive adsorption models are usually applied to consider the simultaneous adsorption of multiple pollutants. For different pollutants, their adsorption types and characteristics can be determined according to empirical methods, and can also be determined through indoor tests or field tests.

In addition, numerous experiments have been conducted on the adsorption properties of different soil types for different solutes using batch and soil column experiments. Xie et al. [79] used batch experiments to study the influence of ammonium ion concentration, temperature, and pH on the adsorption performance of ammonia nitrogen in loess, and obtained the maximum adsorption capacity of ammonium nitrogen in loess by the Langmuir model fitting (72.3 mg/g). Zhao et al. [80] used batch experiments to study the adsorption kinetics and isothermal adsorption of Bisphenol A on three soil surfaces, and analyzed the effects of different soil–water ratios on Bisphenol A adsorption. They showed that the second-order kinetic equation could describe the adsorption process of Bisphenol A. The adsorption rate of Bisphenol A increased till the soil–water ratio reached 1:10; a further increase in the soil–water ratio did not affect it. Therefore, when evaluating the adsorption performance, it is necessary to select a suitable adsorption model for the description so that the results will be more reasonable.

In this section, the adsorption mechanisms and influencing factors of the GM/GCL/AL liner were discussed and the adsorption model of the AL liner was systematically summarized. The anti-pollution performance of the liner can be preliminarily evaluated by the diffusion coefficient. However, the current study on the adsorption characteristics of pollutants through composite liners lacks in the following aspects: (1) The adsorption parameters are currently determined in single-and double-layer liners, and similar studies are required for the three-layer GM/GCL/AL composite liners. (2) In case of triple-layer composite liners, the studies mostly discuss each layer; therefore, systematic and reasonable adsorption models can be established by comprehensively considering the characteristics of each layer and the interface between the liners.

## 5. Degradation of Pollutants through GM/GCL/AL

In contrast to the above mechanism, in the GM/GCL/AL liners, organic pollutants are degraded mainly through microbial degradation in the AL. Even though the properties of the GCL and clay liner are similar, the degradation of pollutants in GCL is negligible owing to its thinness.

Biodegradation is a controlled process that determines the persistence and mobility of organic matter in the soil. Studies have shown that organic pollutants, such as aliphatic and aromatic compounds, can be effectively degraded in the soil [81]. Ren et al. [12,13] found that degradation in municipal solid-waste landfills is the main cause of attenuation of VOCs. The degradation of organic pollutants in soil can effectively reduce landfill leachate pollution into groundwater [82,83]. Therefore, the impact of degradation must also be considered when evaluating the anti-pollution performance of landfill liner systems [84].

The decay of organic matter in the landfill leachate can be divided into two stages: (1) decay in the AL and (2) attenuation in the aquifer. The decay of AL is influenced by ion exchange, precipitation, dissolution, and biodegradation. However, in an aquifer, pollutants migrate with groundwater and are attenuated by filtration and precipitation in the aquifer medium. The organic pollutants in porous media are degraded by microorganisms, as they metabolize the useful compounds, while the compounds that cannot be used in microbial metabolism remain in the leachate. The biodegradation of organic matter is mainly divided into aerobic and anaerobic degradation. To date, numerous experimental and theoretical studies on biodegradation have been published [85,86,87,88]. For the experimental study, most of these studies conducted the tank and column experiment to analyze biodegradation, which is the upper (influent) reservoir concentrations decreased as a function of time even though the influent bag concentrations were kept nearly constant. For most of the VOCs, the concentration decreased faster in the influent reservoir as the soil/water partition coefficient increased. Several factors suggested that VOC degradation was significant even though disinfectants were added to the influent.

For the theoretical study, biodegradation was found to follow a first order reaction; biodegradation is usually expressed by the following equation [88]:$$\frac{\partial C}{\partial t}=-\lambda C$$
where *C* (g/L) is the concentration of pollutants and *λ* is the first-order degradation coefficient, which can be obtained by the following formula:$$\frac{C}{C_{0}}=\frac{1}{2}=e^{-\lambda_{t\;1/2}}$$
where $\lambda =\frac{\ln2}{t_{1/2}}=\frac{0.693}{t_{1/2}}$; *t*_1/2_ is the half-life time of pollutant degradation.

The studies have shown that the anaerobic degradation of organic pollutants occurs in the bottom liner of the landfill, which is the most common degradation mode in soil–water systems. Biological barriers are an effective way of increasing the biodegradation of organic pollutants in barrier systems, thereby reducing the entry of organic pollutants into aquifers.

Li et al. [89] studied the biodegradation of perchloroethylene (PCE) in a biological barrier through batch experiments and analyzed the effects of temperature and pH on it. They showed that the optimum pH for the biodegradation of PCE in the biological barrier was 7, the first-order degradation coefficient of PCE was 0.2489 d^−1^ at 20 °C, but decreased to 0.1233 d^−1^ when the temperature reduced to 12 °C. Careghini et al. [90] conducted soil column and batch experiments and obtained first-order degradation coefficients of benzene, toluene, ethylbenzene, o-xylene, p-xylene, and methyl tert-butyl (0.62, 0.88, 1.67, 0.69, 1.23, and 0.34 d^−1^, respectively). Guillén et al. [91] studied the experimental biodegradation of herbicides’ (mainly composed of picloram) reaction walls and found that the first-order degradation coefficient of herbicides in a biological barrier was 0.924–4.23 d^−1^.

Moreover, Krupa et al. [81] studied the effectiveness of biological barriers for pollutants in landfill leachate through soil column experiments. They showed that the concentration of organic pollutants in the leachate can be significantly reduced through biodegradation, suggesting that the microbial absorption and degradation of organic pollutants in biological barriers is sustainable and effective. Dong et al. [92] showed that the adsorption and biodegradation in the groundwater of landfills were far greater than hydrolysis and photolysis. Wu et al. [93] studied the influence of the degradation half-life time of pollutants in GM/GCL/AL composite liners on the pollutant flux and concentration at the bottom of the liner (Figure 5), which were observed to increase with the extension and increase in the degradation half-life time.

The above studies indicate that biodegradation must be considered when evaluating the antifouling performance of liner systems. Biodegradation is the main form of pollutant degradation and the degradation coefficient can be calculated by above formula. However, there are only a few studies with respect to the degradation of pollutants through composite liners, and most of them involve biological barriers. Such studies need to be conducted to deepen the understanding of the degradation mechanism under the action of different factors.

## 6. Analytical Model for Pollutant Transport

Based on the above summary, the pollutant transport characteristics through the liners can be understood; thus, in order to further evaluate the anti-pollution performance of liners, it is necessary to adopt the numerical methods or analytical solutions for analysis.

Numerical methods [94,95,96] have been previously used to study the transport model of pollutants in composite liners. The finite layer method developed by Smith et al. [97] and the finite difference method proposed by Foose et al. [98] are used to analyze organic pollutant transport through composite liners. El-zein [99] adopted the equivalent boundary method to transform the GM into a certain boundary condition considering both convection and diffusion, and simplified the calculation by showing that the equivalent boundary calculation is more reasonable. Leo [100] studied pollutant transport in heterogeneous media in landfills using the boundary element method. Guyonnet et al. [101] used the Laplace transform numerical inversion method to study the hydrodynamic equivalence between multilayer mineral liners. Cooke and Rowe [102] studied the two-dimensional transport of pollutants in a barrier system at the bottom of a landfill, using the finite element method. The studies show that numerical models effectively simulate pollutant transport through composite liners.

The above numerical models are important for the simulation of pollutant transport problems through composite liners. However, the complexity of the numerical methods often far exceeds the complexity of the relevant data available for practical situations. In such cases, a simplified analytical solution can often be an economical and effective alternative to numerical methods in many ways. The analytical solutions can be used to understand the basic mechanisms of pollutant transport, conduct sensitivity analysis as dimensionless variables, analyze the experimental results, and easily simulate the transport behavior on a larger spatiotemporal scale [103,104].

Foose et al. [105] established a one-dimensional diffusion model of pollutants in a double-layer soil and obtained an analytical solution under a semi-infinite space. Huysmans and Dassargues [106] studied the diffusion of pollutants in layered media by combining the equivalent diffusion coefficient with a one-dimensional diffusion analytical solution for pollutants in homogeneous media. Cleall and Li [107] developed a one-dimensional diffusion model of organic pollutant transport in a composite liner under semi-infinite and finite-thickness bottom boundary conditions and obtained the corresponding analytical solution. Chen et al. [108] studied pollutant transport through a composite liner with defective GM under a high waterhead. They established a convection–diffusion model of heavy metals and organic pollutants through GM/CCL and GCL/AL composite liners and obtained the analytical solutions.

Although several studies propose the convection–diffusion analytical model of solutes through layered media, the analytical model of pollutant transport in liners has been confined to single-layer or double-layer models for a long time. In recent years, studies have been conducted on three-layer liner models. Wu et al. [93] proposed an analytical solution for the steady-state diffusion of degradable organic pollutants through a triple-layer composite liner; however, this analytical solution did not consider the leakage effect of pollutants through GM defects. Xie et al. [15,109] compensated for the previous shortcomings and developed an analytical model that comprehensively considered convection, diffusion, and degradation. The transport state of the obtained model in GM and GCL was steady; however, the transport of organic pollutants in all of the layers was transient. Based on this, Feng et al. [16,17] derived an analytical solution of degradable organic pollutant transport through a GM/GCL/AL liner using the separation of variables method, and accurately described the transient pollutant transport of pollutants in the entire composite liner.

Notably, although the one-dimensional transport model is relatively uncomplicated compared to the multi-dimensional one, it can also describe the transport characteristics to a large extent. Therefore, several detailed studies have been conducted on one-dimensional transport models. Such studies mainly focus on the transport state and transport mechanism. Control equations and boundary conditions were established to obtain the transport model, which was then solved using an algorithm to obtain the analytical solution of the one-dimensional transport model.

### 6.1. Governing Equation of One-Dimensional Transport Model

The form of the governing equation is related to two factors: the transport state (steady or transient) of pollutants in the composite liner and the transport mechanism of pollutants through the GM/GCL/AL liner. The four main transport mechanisms were discussed in the first four sections. The governing equations of the different calculation models from various studies are summarized in this section (Table 5).

As can be seen in Table 5, the governing equation is mainly determined by the transport mechanism through each layer. For instance, Feng et al. [16,17] assumes that the transport of pollutants through the GM layer is by diffusion and convection, and the transport through the GCL and Al layers is by diffusion, convection, and degradation. Then, based on the law of conservation of mass, select the relevant parameters Di, Ki, and λi to establish the governing equation of each layer. Therefore, it is recommended to consider a more consistent transport mechanism according to the actual situation and establish the governing equation.

### 6.2. Boundary Condition of One-Dimensional Transport Model

The boundary conditions include the top and bottom boundary conditions, and the GM/GCL and GCL/AL contact surface continuity conditions. The expression of the boundary conditions differs according to the various transport states. The presently studied transport states include semi-steady, semi-transient, and fully transient. For the GM/GCL/AL, semi-steady denotes steady state transport in the GM and GCL, semi-transient denotes transient state transport in the AL, and fully transient denotes transient state transport in all of the layers.

① The top boundary is the interface between the leachate and GM. The concentration redistribution of pollutants in leachate at the infiltration interface of GM can be described according to different states as follows:

Semi-steady–semi-transient state: $$C_{\mathrm{gm}}\left(0\right)=C_{0}K_{g}$$

Fully transient state:$$C_{\mathrm{gm}}\left(0,t\right)=C_{0}K_{g}$$

② The bottom boundary of the composite liner is assumed to be a zero-concentration (Dirichlet boundary) or zero-flux (Neumann boundary) boundary:$$C_{\mathrm{al}}\left(L_{\mathrm{cl}},t\right)=0\;(\mathrm{Dirichlet}\;\mathrm{boundary}\;\mathrm{condition})$$
$$\frac{\partial C_{\mathrm{al}}\left(L_{\mathrm{cl}},t\right)}{\partial z}=0\;(\mathrm{Neumann}\;\mathrm{boundary}\;\mathrm{condition})$$
where L_cl_ is the total thickness of the triple-layer liner.

③ Continuity interface conditions include pollutant concentration continuity and continuous flux conditions.

Semi-steady–semi-transient state:$$\frac{C_{gm}\left(L_{gm}\right)}{K_{g}^{\prime }}=C_{gcl}\left(L_{gm}\right)$$
$$D_{gm}\frac{\partial C_{gmb}\left(L_{gm}\right)}{\partial z}=n_{gcl}D_{gcl}\frac{\partial C_{gcl}\left(L_{gm}\right)}{\partial z}$$
$$C_{gcl}\left(L_{gm}+L_{gcl}\right)=C_{al}\left(L_{gm}+L_{gcl},t\right)$$
$$n_{gcl}D_{gcl}\frac{\partial C_{gcl}\left(L_{gm}+L_{gcl}\right)}{\partial z}=n_{al}D_{al}\frac{\partial C_{al}\left(L_{gm}+L_{gcl},t\right)}{\partial z}$$
where *K′_g_* is the distribution coefficient between GM and GCL.

Fully transient state:$$\frac{C_{gm}\left(L_{gmb},t\right)}{K_{g}^{\prime }}=C_{gcl}\left(L_{gm},t\right)$$
$$D_{gmb}\frac{\partial C_{gmb}\left(L_{gmb},t\right)}{\partial z}=n_{gcl}D_{gcl}\frac{\partial C_{gcl}\left(L_{gmb},t\right)}{\partial z}\;\mathrm{or}$$
$$v_{a}\frac{C_{\mathrm{gm}}\left(L_{\mathrm{gm}},t\right)}{K_{g}^{\prime }}-D_{\mathrm{gm}}\frac{\partial C_{\mathrm{gm}}\left(L_{\mathrm{gm}},t\right)}{\partial z}=n_{\mathrm{gl}}v_{\mathrm{gcl}}C_{\mathrm{gcl}}\left(L_{\mathrm{gm}},t\right)-n_{\mathrm{gcl}}D_{\mathrm{gcl}}\frac{\partial C_{\mathrm{gcl}}\left(L_{\mathrm{gm}},t\right)}{\partial z}$$
$$C_{gcl}\left(L_{gm}+L_{gcl},t\right)=C_{al}\left(L_{gm}+L_{gcl},t\right)$$
$$n_{gcl}D_{gcl}\frac{\partial C_{gcl}\left(L_{gm}+L_{gcl},t\right)}{\partial z}=n_{al}D_{al}\frac{\partial C_{al}\left(L_{gm}+L_{gcl},t\right)}{\partial z}$$

Additionally, the initial conditions are:$$\left[\begin{array}{c} C_{\mathrm{gm}}\left(z,0\right)\\ C_{\mathrm{gcl}}\left(z,0\right)\\ C_{\mathrm{al}}\left(z,0\right) \end{array}\right]=0$$

### 6.3. Analytical Solution of One-Dimensional Transport Model

The one-dimensional transport model can be solved by combining the governing equations and boundary conditions. However, the model cannot be solved directly because the governing equations are ordinary differential equations, and the boundary conditions have several parameters. Therefore, analytical solution methods have been developed. The currently available solution methods for the one-dimensional transport model of GM/GCL/AL are mainly divided into the following two types:

① First, the governing equations and boundary conditions are generalized to make them applicable to each layer of the liner; then, the problem is decomposed into two sub-problems according to the transport state, and different conditions are substituted into the sub-problems to solve them individually [105,106]. When convection, diffusion, and degradation are comprehensively considered and the transport state is fully transient, it can be expressed as:

Generalized governing equation:$$\frac{\partial C_{i}\left(z,t\right)}{\partial t}=\frac{D_{i}}{R_{d,i}}\frac{\partial^{2}C_{i}\left(z,t\right)}{\partial z^{2}}-\frac{v_{i}}{R_{d,i}}\frac{\partial C_{i}\left(z,t\right)}{\partial z}-\lambda_{i}C_{i}\left(z,t\right),i=1,2,3$$

Generalized continuous condition:$${\left.C_{i}\left(z,t\right)\right|}_{z=L_{i}}={\left.C_{i+1}\left(z,t\right)\right|}_{z=L_{i}},i=1,2$$
$${\left.n_{i}D_{i}\frac{\partial C_{i}\left(z,t\right)}{\partial z}\right|}_{z=L_{i}}={\left.n_{i+1}D_{i+1}\frac{\partial C_{i+1}\left(z,t\right)}{\partial z}\right|}_{z=L_{i}},i=1,2$$

Decomposition subproblem:$$C_{i}\left(z,t\right)=w_{i}\left(z,t\right)+u_{i}\left(z\right)C_{0},i=1,2,3$$

Sub problem 1:$\frac{\partial w_{i}\left(z,t\right)}{\partial t}=\frac{D_{i}}{R_{d,i}}\frac{\partial^{2}w_{i}\left(z,t\right)}{\partial z^{2}}-\frac{v_{i}}{R_{d,i}}\frac{\partial w_{i}\left(z,t\right)}{\partial z}-\lambda_{i}w_{i}\left(z,t\right),i=1,2,3$

Sub problem 2:$\frac{D_{i}}{R_{d,i}}\frac{d^{2}u_{i}\left(z\right)}{dz^{2}}-\frac{v_{i}}{R_{d,i}}\frac{du_{i}\left(z\right)}{dz}-\lambda_{i}u_{i}\left(z\right)=0,i=1,2,3$ where *i* = 1, 2, and 3 represent the GM, GCL, and AL, respectively;

② Governing equations of each layer of the liner are solved separately to obtain the general solution with parameters, which is then substituted into the different boundary conditions to obtain the parameters and their corresponding analytical solution [15,109]. Considering the solving process of Xie et al. [15] as an example:

General solution of GM: $C_{gm}\left(z\right)=A_{1}+A_{2}\exp\left(\frac{v_{a}z}{D_{gmb}}\right)$

General solution of GCL: $C_{gcl}\left(z\right)=A_{3}+A_{4}\exp\left(\frac{v_{a}z}{n_{gcl}D_{gcl}}\right)$

Notably, AL has two different bottom boundary conditions; therefore, the solution may differ depending on them. The Laplace change method is often used to solve the problem for the Neumann boundary condition, while the separation of variables method is often used for the Dirichlet boundary condition, which is the same as the ① solution, where the two sub-functions are solved individually. Considering the solving process of Xie et al. [15] as an example.

For the Neumann boundary condition:$$\bar{C_{al}\left(z,p\right)}=\int_{0}^{\infty }=e^{-pt}C_{al}\left(z,t\right)dt$$
where *p* is the Laplace variation parameter.

The governing equation of AL under such conditions is:$$p\bar{C_{al}\left(z,p\right)}-C_{i}=\frac{D_{al}}{R_{d}}\frac{\partial^{2}\bar{C_{al}\left(z,p\right)}}{\partial z^{2}}-\frac{v_{al}}{R_{d}}\frac{\partial \bar{C_{al}\left(z,p\right)}}{\partial z}-\lambda \bar{C_{al}\left(z,p\right)}$$

For the Dirichlet boundary condition: $C_{al}\left(z,t\right)=\theta \left(z,t\right)+\phi \left(z\right)$ where *θ(z,t)* and *Φ(z)* are the variables satisfying the homogeneous and non-homogeneous differential equation, respectively.

Sub problem 1: $\frac{\partial \theta \left(z,t\right)}{\partial t}=\frac{D_{al}}{R_{d}}\frac{\partial^{2}\theta \left(z,t\right)}{\partial z^{2}}-\frac{v_{al}}{R_{d}}\frac{\partial \theta \left(z,t\right)}{\partial z}-\lambda \theta \left(z,t\right)$

Sub problem 2: $\frac{D_{al}}{R_{d}}\frac{\partial^{2}\phi \left(z\right)}{\partial z^{2}}-\frac{v_{al}}{R_{d}}\frac{\partial \phi \left(z\right)}{\partial z}-\lambda \phi \left(z\right)=0$

Summarizing the previous studies on the analytical solution for the one-dimensional transport model of GM/GCL/AL, it is inferred that the transport model is generally established through the control equations and boundary conditions and then solved to obtain the analytical solution.

In conclusion, the analytical solution is an effective way to evaluate the anti-pollution performance of composite liners. The key point of this method is to determine the governing equations and boundary conditions of the pollutant transport; thus, the readers can select different transport mechanisms’ combinations according to actual conditions to establish the governing equations and select boundary conditions based on the transport state. The analytical solution obtained by solving the sub-problem can calculate the pollutant concentration and flux at the bottom of the liners, and then evaluate the anti-pollution performance of the liners.

However, the above mentioned studies are lacking in the following aspects under different circumstances: (1) Owing to its complexity, the studies of transport models based on the finite element method are currently at the bottleneck stage. Thus, it is necessary to develop a simplified and reliable numerical simulation method suitable for the actual situation. (2) The transport models established by the analytical method are mostly one-dimensional, and can be expanded into two-dimensional and three-dimensional problems of pollutant transport in composite liners. (3) The analytical solution method of the transport model must be studied in detail. (4) To simulate the actual conditions of the landfill site, it might be necessary to study the transport model considering more comprehensive transport effects and states.

## 7. Breakthrough Time

Based on the transport model mentioned before, the expression of breakthrough time can be further derived. The breakthrough time of pollutants is one of the important indexes to evaluate the anti-pollution performance of liners, which means the transport time required for the pollutant concentration in the leachate (*C*_0_) to reach a certain level (*C_b_*) at the bottom of the liner system, where it becomes the breakthrough concentration. Most countries refer to the contaminant threshold concentration in water quality standards as the breakthrough concentration value.

To study the different indicator pollutants based on different breakthrough concentration standards, Foose [110] used cadmium and dichlorotoluene for the steady-state calculation of transport–dispersion, and the breakthrough time was judged by the maximum contaminant level in the American drinking water standard. Xie et al. [111] used cadmium and benzene to calculate the analytical solution of one-dimensional transport–dispersion, which was evaluated by the pollutant concentration in the national primary drinking water regulations. Zhan et al. [112] used a chloride ion centrifuge to simulate the transport–dispersion of breakthrough pollutants under a high waterhead and selected 10% of the initial concentration to determine the breakthrough time. Zhu et al. [110] established a one-dimensional convection–dispersion finite element model using the organic pollutant Chemical Oxygen Demand (COD) as an indicator, and calculated the breakthrough time based on the concentration standard of COD in China’s groundwater and surface water environment.

The above studies suggest that the calculation methods of breakthrough time mainly include the finite element and analytical solution methods. These two methods help to establish the transport model of pollutants in the composite liner, calculate the pollutant concentration at different moments according to the solution of the model, and finally determine the breakthrough time according to the corresponding concentrations of the breakthrough standard. Considering these aspects, mathematical expressions have been developed for the breakthrough time of pollutants in different composite liners under different effects (Table 6).

As observed from the above table, the breakthrough time of GM/GCL/AL was mainly calculated using the pollutant concentration index of the bottom layer (AL), and the calculation was derived from the analytical solution of the transport model. Xie et al. [15] provided a dimensionless expression for parameters related to the relative concentration (*C_N_*) and relative flux (*F_N_*) under semi-infinite boundary conditions and drew a dimensionless design curve of *C_N_* and *F_N_* for *T_R_*.

The dimensionless parameters in the expression were defined as follows:$$K=\frac{K_{g}}{K_{g}^{\prime }}$$
$$P_{1}=\frac{v_{al}\left(z-L_{gcl}-L_{gmb}\right)}{D_{al}}$$
$$P_{2}=\frac{\eta v_{al}}{D_{al}}$$
$$T_{R}=\frac{D_{al}t}{R_{d}{\left(z-L_{gcl}-L_{gm}\right)}^{2}}$$
$$Q=\frac{R_{d}\lambda {\left(z-L_{gcl}-L_{gm}\right)}^{2}}{D_{al}}$$
where *K* is the ratio of the distribution coefficient; *P_1_* represents the relative ratio of convection and diffusion in AL; *P_2_* reflects the transport characteristics of pollutants in the GM and GCL; and *Q* represents the relative ratio of degradation and diffusion in AL. Additionally, the value of time factor *T_R_* was determined by two dimensionless Peclet numbers. The specific relationship is shown in Figure 6 and Figure 7.

Furthermore, the factors influencing the breakthrough time have also been discussed. Zhu et al. [18] demonstrated that the initial concentration of pollutants is one of the influencing factors. Different initial concentrations lead to different pollutant concentrations during chemical diffusion. Therefore, a higher initial concentration generally denotes a greater concentration gradient and a shorter breakthrough time. Additionally, adsorption hinders pollutant transport, resulting in a prolonged breakthrough time; therefore, the retardation factors of the liner also affect the degree of extension of the breakthrough time. Chen [114] found that the permeability coefficient and retardation factor are the key factors affecting the breakthrough time, while the hydrodynamic dispersion coefficient and waterhead height have a minor influence on it.

In summary, the breakthrough time is an important indicator for evaluating the long-term performance of a composite liner and predicting its service life. It is determined using indicator pollutants and the corresponding breakthrough concentration standards. Although there are many methods to calculate the breakthrough time, it is simpler and more effective to deduce and calculate the breakthrough time through the analytical solution. Comparing the pollutant concentration at the bottom of the liner with the concentration specified in the standard, it is proved that the GM/GCL/AL composite liners have a good anti-pollution performance.

This section summarized the calculation formulas of breakthrough times under different conditions and discussed its influencing factors. However, the available studies lack in the following aspects: (1) Breakthrough time is mainly calculated from the transport model, which lacks independent and systematic research. (2) The factors influencing the breakthrough time of composite liners are relatively single, and the effects of multiple factors and their combinations on the breakthrough time must be considered. (3) Few studies have been conducted on the breakthrough time of GM/GCL/AL.

## 8. Summary and Prospects

The research improvements and developments of GM/GCL/AL triple-layer composite liners in recent decades were summarized and the anti-pollution performance of the liners was comprehensively evaluated in this paper. The main conclusions are as follows.

(1)The convection of pollutants through GM/GCL/AL is mainly leakage through GM defects, which can be evaluated by calculating the leakage rate. It is recommended to determine the type of defect according to the actual situation on site and calculate the leakage rate according to the formula. Additionally, factors such as the interface transmissivity coefficient and properties of the underlying medium also affect convection, which are also necessary to be considered. However, the studies on pollutant convection through composite liners have mainly focused on single defects rather than multiple or combination of defects, and lack systematic research on the GM/GCL/AL characteristics;(2)The diffusion of pollutants through GM/GCL/AL mainly consider the transport of organic pollutants. The diffusion coefficient is the parameter to evaluate the diffusion characteristics, and it is recommended to measure it by an indoor diffusion test. In addition, the chemical properties of the pollutants and the consolidation characteristics of the soil are also important factors, which should be considered in the actual situation. In future research, the diffusion behavior of other pollutants in the composite liner could be studied to establish the diffusion coefficient database of different pollutants through the composite liner;(3)The adsorption of pollutants through the GM/GCL/AL was mainly evaluated by the adsorption coefficient. The adsorption coefficient is recommended to be obtained through an indoor adsorption test. In addition, the adsorption model is also the basis to evaluate the adsorption characteristics. It is recommended to select a suitable adsorption model according to the actual situation; the Langmuir model and the two-stage adsorption model are the more common and reasonable models in the current study. However, current research on triple-layer liners is limited, and the interaction and interface characteristics between liners are not comprehensively considered;(4)The degradation of organic pollutants in soil can effectively reduce landfill leachate pollution into groundwater, so it is suggested to consider the impact of degradation when evaluating the anti-pollution performance of landfill liner systems. Biodegradation is the main form of pollutant degradation, and the degradation coefficient can be calculated by the summarized formula. However, it is necessary to strengthen the study of the degradation characteristics of pollutants in the triple-layer GM/GCL/AL composite liners and deepen the research on degradation mechanisms under different conditions;(5)The research methods for pollutant transport models mainly include numerical methods and analytical method. An analytical solution is more recommended because of its simplicity and effectiveness, and this study mainly discussed the analytical solution of a one-dimensional pollutant transport model. Determining the governing equations and boundary conditions are the keys to solving the problem. It is suggested to establish appropriate governing equations according to the transport action, select boundary conditions based on the transport state, and then obtain analytical solutions by the subproblem method. However, in the current studies, the available transport models lack simplified and practical numerical methods, with few studies on multi-dimensional transport models. Furthermore, the simplicity of the transport mechanisms and states could be considered;(6)The breakthrough time of pollutants is one of the important indexes to evaluate the anti-pollution performance of liners. There are many methods to calculate the breakthrough time, and in this paper, it is recommended to use the analytical solution of the transport model for derivation and calculation. By comparing the pollutant concentration at the bottom of the liner with the concentration specified in the standard, it is proved that the GM/GCL/AL composite liners have good anti-pollution performance. However, there is still a lack of separate and systematic studies on breakthrough times, such as the use of a single evaluation index of indicator pollutants. Thus, studies need to be conducted on the coupling effect of multiple factors.

## Figures and Tables

**Figure 1 membranes-12-00922-f001:**
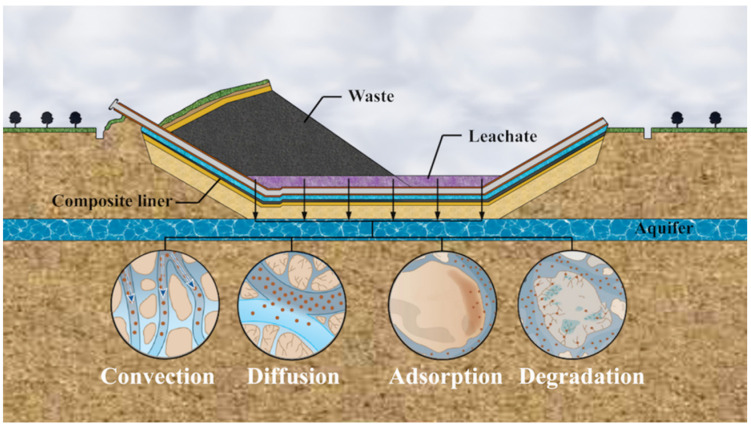
Schematic diagram of GM/GCL/AL (geomembrane/geosynthetic clay liner/attenuation layer) triple-layer liners.

**Figure 2 membranes-12-00922-f002:**
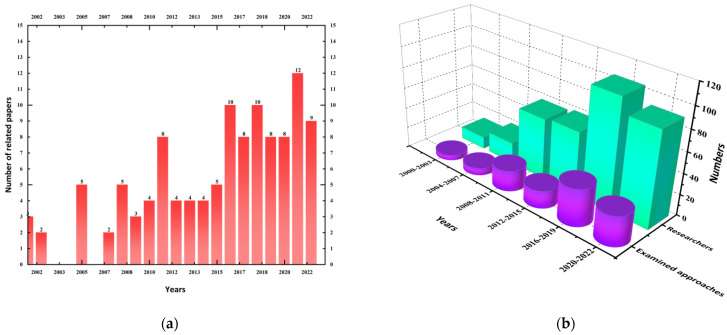
Research trend and progress in recent decades: (**a**) The number of related papers for last 20 years; (**b**) The number of researchers and examined approaches; (**c**) The number of related papers in different countries; (**d**) The percentage of pollution-sealing approaches.

**Figure 3 membranes-12-00922-f003:**
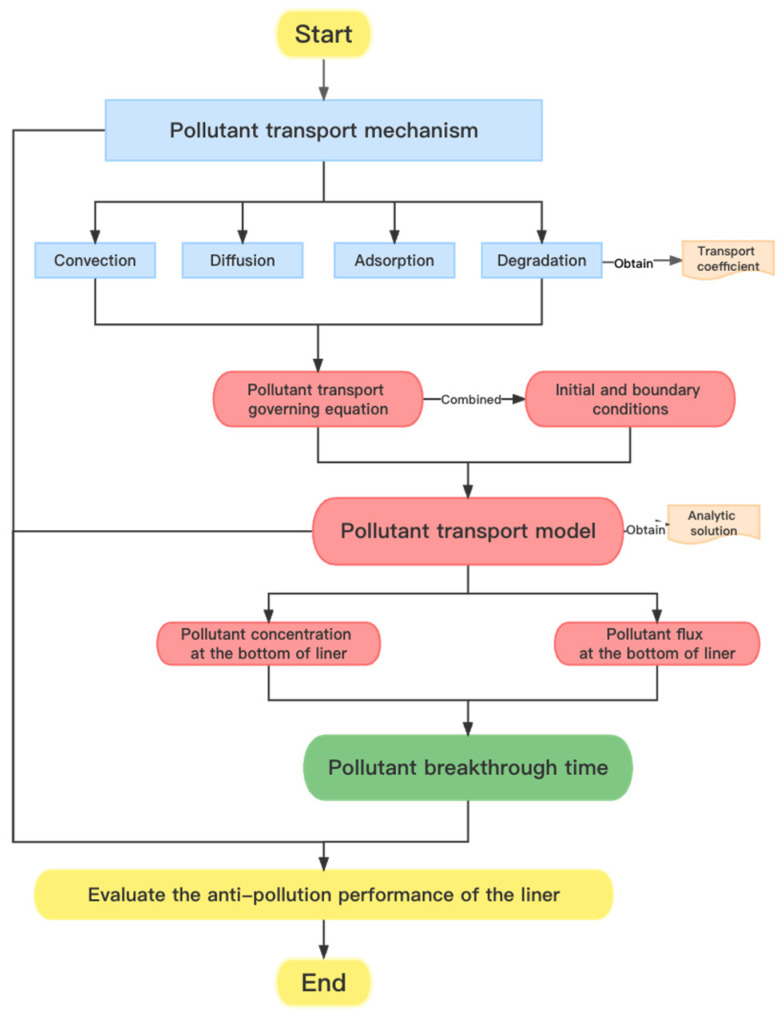
Flow chart of the research process.

**Figure 4 membranes-12-00922-f004:**
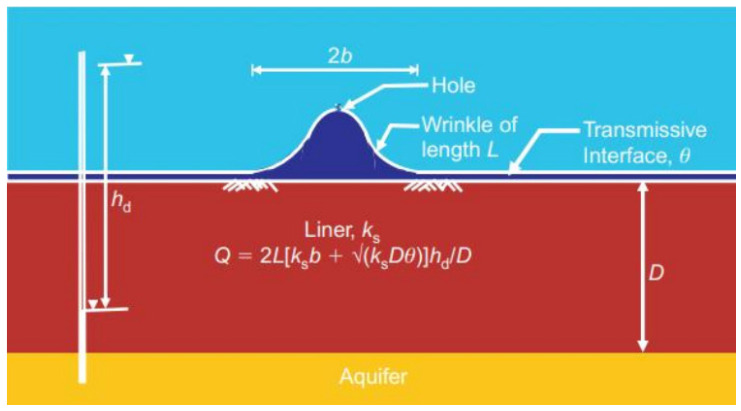
Schematic diagram of pollutant leakage from the hole connected with the wrinkle. Redrawn from Ref. [35].

**Figure 5 membranes-12-00922-f005:**
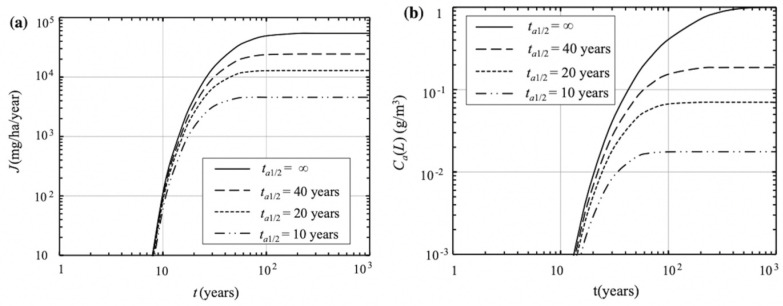
Effect of degradation on pollutant flux (**a**) and concentration (**b**) at the bottom of GM/GCL/AL liners. Reprinted with permission from Ref. [93], Copyright 2016 Springer.

**Figure 6 membranes-12-00922-f006:**
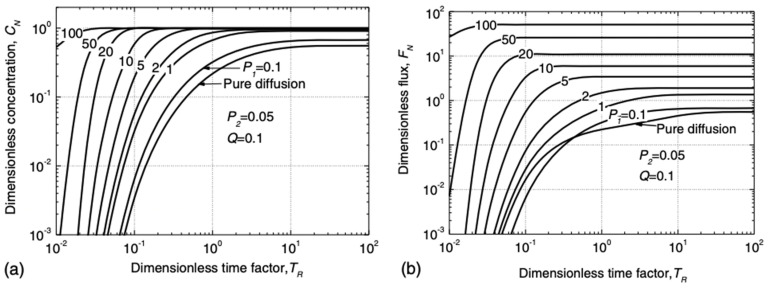
Variation curve of *C_N_* and *F_N_* of pollutant with *P*_1_ (*P*_2_ = 0.05, *Q* = 0.1): (**a**) dimensionless concentration *C_N_*; (**b**) dimensionless flux *F_N_*. Reprinted with permission from Ref. [15], Copyright 2018 ASCE Library.

**Figure 7 membranes-12-00922-f007:**
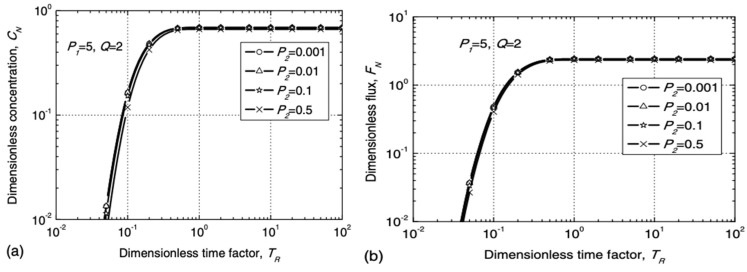
Variation curve of *C_N_* and *F_N_* of pollutant with *P*_2_ (*P*_1_ = 5, Q = 2): (**a**) dimensionless concentration *C_N_*; (**b**) dimensionless flux *F_N_*. Reprinted with permission from Ref. [15], Copyright 2018 ASCE Library.

**Table 1 membranes-12-00922-t001:** Key evaluation indicators and research contents.

Evaluation Indicator	Main Content	Evaluation Parameters	Main Research Methods	Important References
Pollutants’ transport mechanism	Convection	Leakage rate	Experimental research; Empirical formula	[3,4,5]
	Diffusion	Diffusion coefficient	Experimental research; Empirical formula	[6,7,8]
	Adsorption	Adsorption coefficient	Experimental research; Adsorption model	[9,10,11]
	Degradation	Degradation coefficient	Experimental research; Empirical formula	[12,13,14]
Pollutants’ transport model	Governing equation	Pollutant concentration and flux	Analytical solution	[15,16,17]
	Boundary conditions			
Pollutants’ breakthrough time	Breakthrough time	Indicator pollutant concentration	Transport model derivation	[18,19,20]

**Table 2 membranes-12-00922-t002:** Calculation method of leakage rate of different defects.

Defects Type	Applicable Conditions	Calculation Method
Circular hole [3]	Good contact; contact interface conditions: log(*K_i_*/*K_s_*) = 1	$Q_{c}=F_{c}K_{s}h_{t}r$ $F_{c}=4+3.35\frac{r}{L_{s}}$
Circular hole [3]	Soil composite liner is relatively thick (thickness ≥ 61 cm); contact interface condition: log(*K_i_*/*K_s_*) > 4.5	$Q_{c,G}=\beta_{c}\left[1+0.1{\left(\frac{h_{w}}{L_{s}}\right)}^{0.95}\right]a^{0.1}h_{w}^{0.9}K_{s}^{0.7}$
Circular hole [4]	Soil composite liner is relatively thick (thickness ≥ 61 cm); contact interface condition: log(*K_i_*/*K_s_*) < 4.5	Numerical model method is recommended
Circular hole [4]	GCL composite liner; contact interface condition: log(*K_i_*/*K_s_*) > 4	$Q_{c,R}=\pi K_{s}\left(r^{2}\frac{h_{t}}{L_{s}}+2\frac{h_{t}}{L_{s}}\Delta_{1}+2\frac{h_{t}}{L_{s}}\Delta_{2}-2\frac{h_{w}}{L_{s}}\Delta_{2}\right)$
Circular hole [4]	GCL composite liner; contact interface condition: log(*K_i_*/*K_s_*) < 4	$Q_{a}=2.85Q_{R}{\left[\log\left(\frac{K_{i}}{K_{s}}\right)\right]}^{-0.73}$
Rectangular hole [5]	Good contact; contact interface conditions: log(*K_i_*/*K_s_*) = 1	$Q_{l}=F_{l}K_{s}h_{t}$ $F_{l}=\frac{1}{0.52-0.76\log\left(\frac{w}{L_{s}}\right)}$
Rectangular hole [5]	Soil composite liner is relatively thick (thickness ≥ 61 cm); contact interface condition: log(*K_i_*/*K_s_*) > 4.5	$Q_{l}=F_{l}K_{s}h_{t}$ $F_{l}=\frac{2}{L_{s}}\left[\frac{w}{2}+\sqrt{\frac{L_{s}T}{K_{s}}}\right]$
Rectangular hole [21]	Soil composite liner is relatively thick (thickness ≥ 61 cm); contact interface condition: log(*K_i_*/*K_s_*) < 4.5	Numerical model method is recommended
Rectangular hole [21]	GCL composite liner; contact interface condition: log(*K_i_*/*K_s_*) > 4	$Q_{a}=2.85Q_{R}{\left[\log\left(\frac{K_{i}}{K_{s}}\right)\right]}^{-0.73}$
Rectangular hole [22]	GCL composite liner; contact interface condition: log(*K_i_*/*K_s_*) < 4	$Q_{a}=2.85Q_{R}{\left[\log\left(\frac{K_{i}}{K_{s}}\right)\right]}^{-0.73}$

Notes: *F_c_* is the dimensionless parameter of the circular hole defect; *K_s_* (m/s) is the saturated permeability coefficient of the soil liner; *h_t_* (m) is the water head difference in the composite liner; r (m) is the radius of the defect hole; *L_s_* is the thickness of the underlying liner (CCL, GCL, or Al); hw (m) is the water head above the GM layer; *a* (m^2^) is the area of the defective hole; the value of *β_c_* depends on the condition of the contact interface, *β_c_* = 0.21 denotes good contact and *β_c_* = 1.15 denotes poor contact; *K_s_* (m^2^/s) is the saturated permeability coefficient of the underlying liner; Δ_1_ and Δ_2_ are expressions related to the Bessel function, defect radius (*r*), and hydration radius (*R*); *Qa* is the adjusted leakage rate of the GCL composite liner in the one-dimensional case, *Q_a_* = *Q_c_*, *R*; *F_l_* is the dimensionless parameter of the rectangular hole defect; and w is the width of the wrinkle.

**Table 3 membranes-12-00922-t003:** Empirical formula of the partition coefficient of organic pollutants through HDPE GM [63].

Influencing Factors	Empirical Formula	Fitting Degree R^2^
Log K_ow_	Log S_gf_ =−1.1523 + 1.2355 Log K_ow_	0.97
M_W_	Oxide: Log S_gf_ =−3.8883 + 10.0363 Log M_w_	0.81
	Chloride: Log S_gf_ =−2.0467 + 10.0305 Log M_w_	0.94
	Aromatic: Log S_gf_ =−0.0776 + 10.0322 Log M_w_	0.95
	Aliphatic: Log S_gf_ =−0.1107 + 10.0442 Log M_w_	0.91

**Table 4 membranes-12-00922-t004:** Classical adsorption models of various adsorption modes.

Adsorption Modes	Adsorption Models	Model Expressions
Equilibrium adsorption	Henry model [10]	$S=K_{d}C$
	Freundlich model [10]	$S=K_{F}C^{1/n}$
	Langmuir model [11]	$S=\frac{S_{\max}K_{L}C}{1+K_{L}C}$
	Two-stage adsorption model [74]	$S=KC,\;C<C_{\max}$ $S=S_{\max},\;C⩾\;C_{\max}$
	Dubinin-Radushkevich (D-R) model [70]	$\ln S=\ln S_{\max}-k\varepsilon^{2}$ $\varepsilon =RT\ln\left(1+\frac{1}{C}\right)$ $E=-\frac{1}{\sqrt{2k}}$
Non-equilibrium adsorption	Linear irreversible model [71]	$\frac{dS}{dt}=\lambda_{1}C$
	Linear reversible model [71]	$\frac{dS}{dt}=\lambda_{2}C-\lambda_{3}S$
	Freundlich model [72]	$\frac{dS}{dt}=\lambda_{4}C^{N}-\lambda_{5}S$
	Langmuir model [72]	$\frac{dS}{dt}=\lambda_{6}C\left(S_{\max}-S\right)-\lambda_{7}S$
	Quasi-first-order kinetic model [73]	$\frac{dS}{dt}=k_{1}\left(S_{e}-S\right)$
	Quasi-second-order kinetic model [75]	$\frac{dS}{dt}=k_{2}{\left(S_{e}-S\right)}^{2}$
	Intraparticle diffusion kinetic model [76]	$S=k_{\mathrm{int}}t{\;}^{0.5}+S_{0}$
Desorption	First-order kinetic model [77]	$S_{t}=S_{0}e^{-k_{del}\;t}$
Competitive adsorption	Sheindorf–Rebhun–Sheintuch (SRS) model [78]	$S_{i}=K_{i}C_{i}{\left[\overset{l}{\underset{j=1}{\sum }}\alpha_{i,j}C_{j}\right]}^{N_{i}-1}$
	Langmuir model [78]	$S_{i}=\frac{K_{i}C_{i}S_{\max}}{1+\sum_{j}K_{j}C_{j}}$

Notes: ① Equilibrium adsorption: *K_d_* is the partition coefficient, *K_F_* and *N* are Freundlich model constants, *S* (g) represents the mass of solute adsorbed per unit mass of soil at equilibrium, *S_max_* represents the maximum adsorption capacity, *K_L_* is the Langmuir model constant, *K* is the partition coefficient of static equilibrium adsorption before saturation, *C* (g/L) is the solution concentration, *C_max_* (g/L) is the critical solution concentration at the maximum adsorption capacity, *k* is the parameter of the D-R model, *ε* is the Polanyi potential, *R* is the ideal gas constant, and *t* (°C) is the thermodynamic temperature. ② Non-equilibrium adsorption: *λ*_1_ is the reaction rate constant; *λ*_2_, *λ*_4_, and *λ*_6_ are the first-order positive reaction rate constants; *λ*_3_, *λ*_5_, and *λ*_7_ are the first-order negative reaction rate constants; *k*_1_ is the quasi-first-order kinetic adsorption rate constant; *k*_2_ is the quasi-second-order kinetic adsorption rate constant; k_int_ is the intraparticle diffusion rate constant; *S*_0_ is the intercept, which is related to the thickness of the boundary layer. ③ Desorption: *S_t_* (g) is the residual amount of solute on the adsorbent at time *t*, *S*_0_ (g) is the mass of the adsorbed solute at the beginning of desorption, *K_del_* is the first-order kinetic adsorption rate constant, and *t* is the desorption time. ④ Competitive adsorption: subscripts *i* and *j* denote the *i* and *j* solutes, respectively; *l* is the total number of solutes, *K_i_* and *Kj* are the adsorption model constants of a single adsorption system when the *i* and *j* solutes exist alone, respectively, *C_i_* and *C_j_* (g/L) are the concentrations of solutes *i* and *j*, respectively, and *α_i,j_* is the dimensionless competitive adsorption parameter of component *i* in the presence of component *j*.

**Table 5 membranes-12-00922-t005:** Governing equations of the different transport models.

Transport Mechanism and State	Schematic Diagram of Transport Model	Governing Equations
GM/GCL: diffusion (steady state) AL: diffusion (transient state) Reprinted with permission from Ref. [109], Copyright 2013 Springer	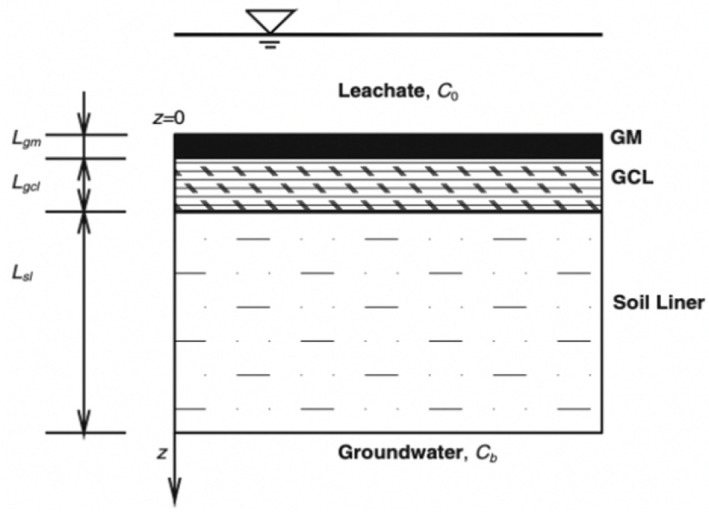	$D_{\mathrm{gm}}\frac{d^{2}C_{\mathrm{gm}}\left(z\right)}{dz^{2}}=0$ $\begin{array}{c} D_{\mathrm{gcl}}\frac{d^{2}C_{\mathrm{gcl}}\left(z\right)}{dz^{2}}=0 \end{array}$ $\frac{\partial C_{\mathrm{al}}\left(z,t\right)}{\partial t}=\frac{D_{\mathrm{al}}}{R_{d,\mathrm{al}}}\frac{\partial^{2}C_{\mathrm{al}}\left(z,t\right)}{\partial z^{2}}$
GM:diffusion (steady state) GCL/AL: diffusion + degradation (transient state) Reprinted with permission from Ref. [15], Copyright 2016 Springer	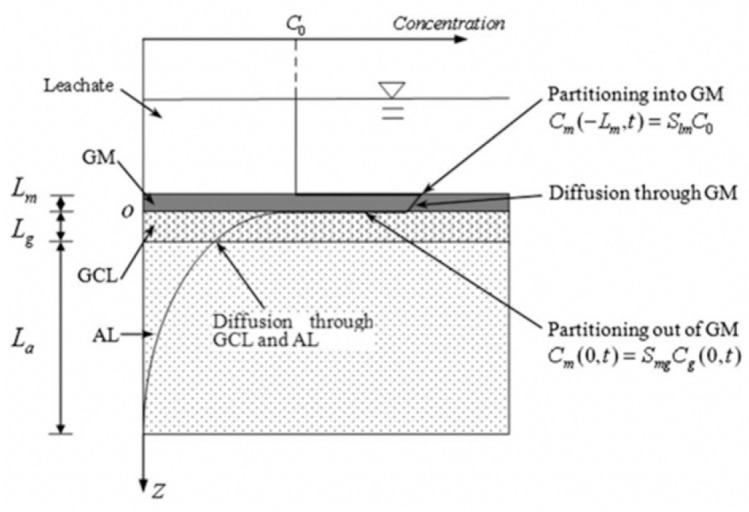	$D_{\mathrm{gm}}\frac{d^{2}C_{\mathrm{gm}}\left(z\right)}{dz^{2}}=0$ $\frac{\partial C_{\mathrm{gcl}}\left(z,t\right)}{\partial t}=\frac{D_{\mathrm{gcl}}}{R_{d,\mathrm{gcl}}}\frac{\partial^{2}C_{\mathrm{gcl}}\left(z,t\right)}{\partial z^{2}}-\lambda_{\mathrm{gcl}}C_{\mathrm{gcl}}\left(z,t\right)$ $\frac{\partial C_{al}\left(z,t\right)}{\partial t}=\frac{D_{al}}{R_{d,al}}\frac{\partial^{2}C_{al}\left(z,t\right)}{\partial z^{2}}-\lambda C_{al}\left(z,t\right)$
GM/GCL: diffusion + convection (steady state) AL: diffusion + convection + degradation (transient state) Reprinted with permission from Ref. [104], Copyright 2018 ASCE Library	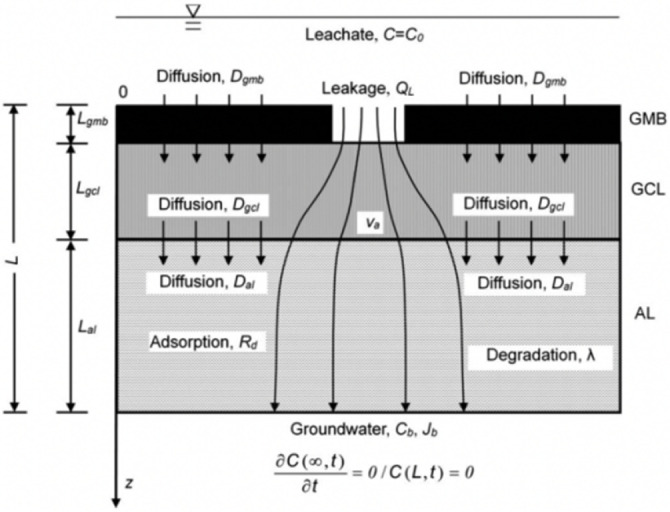	$D_{gm}\frac{\partial^{2}C_{gm}}{\partial z^{2}}-v_{a}\frac{\partial C_{\mathrm{gm}}}{\partial z}=0$ $D_{gcl}\frac{\partial^{2}C_{\mathrm{gcl}}}{\partial z^{2}}-\frac{v_{a}}{n_{gcl}}\frac{\partial C_{\mathrm{gcl}}}{\partial z}=0$ $\frac{\partial C_{al}\left(z,t\right)}{\partial t}=\frac{D_{al}}{R_{d,al}}\frac{\partial^{2}C_{al}\left(z,t\right)}{\partial z^{2}}-\frac{v_{al}}{R_{d,al}}\frac{\partial C_{al}\left(z,t\right)}{\partial z}-\lambda C_{al}\left(z,t\right)$
GM/GCL/AL: diffusion + convection (transient) Reprinted with permission from Ref. [16], Copyright 2018 Elsevier	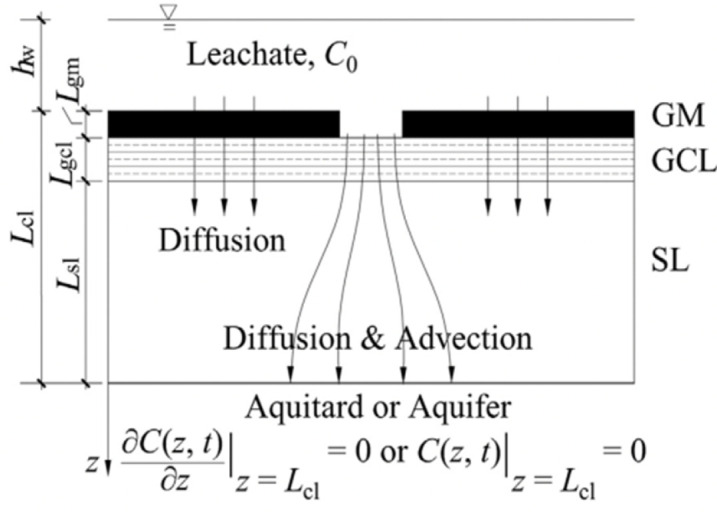	$\frac{\partial C_{\mathrm{gm}}\left(z,t\right)}{\partial t}=D_{\mathrm{gm}}\frac{\partial^{2}C_{\mathrm{gm}}\left(z,t\right)}{\partial z^{2}}-\frac{v_{a}}{K_{g}}\frac{\partial C_{\mathrm{gm}}\left(z,t\right)}{\partial z}$ $\frac{\partial C_{\mathrm{gcl}}\left(z,t\right)}{\partial t}=\frac{D_{\mathrm{gcl}}}{R_{d,\mathrm{gcl}}}\frac{\partial^{2}C_{\mathrm{gcl}}\left(z,t\right)}{\partial z^{2}}-\frac{v_{\mathrm{gcl}}}{R_{d,\mathrm{gcl}}}\frac{\partial C_{\mathrm{gcl}}\left(z,t\right)}{\partial z}$ $\frac{\partial C_{al}\left(z,t\right)}{\partial t}=\frac{D_{al}}{R_{d,al}}\frac{\partial^{2}C_{al}\left(z,t\right)}{\partial z^{2}}-\frac{v_{al}}{R_{d,al}}\frac{\partial C_{al}\left(z,t\right)}{\partial z}$
GM: diffusion + convection (transient) GCL/Al: diffusion + convection + degradation (transient) Reprinted with the permission from Ref. [17], Copyright 2019 Elsevier	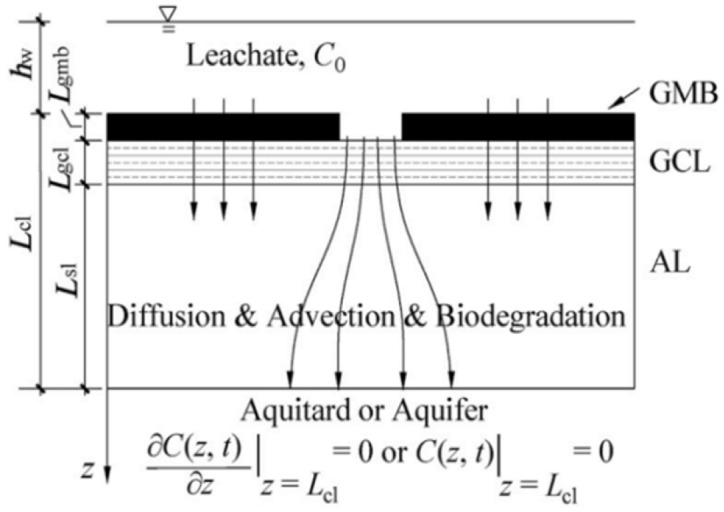	$\frac{\partial C_{\mathrm{gm}}\left(z,t\right)}{\partial t}=D_{\mathrm{gm}}\frac{\partial^{2}C_{\mathrm{gm}}\left(z,t\right)}{\partial z^{2}}-v_{a}\frac{\partial C_{\mathrm{gm}}\left(z,t\right)}{\partial z}$ $\frac{\partial C_{\mathrm{gcl}}\left(z,t\right)}{\partial t}=\frac{D_{\mathrm{gcl}}}{R_{d,\mathrm{gcl}}}\frac{\partial^{2}C_{\mathrm{gcl}}\left(z,t\right)}{\partial z^{2}}-\frac{\nu_{\mathrm{gcl}}}{R_{d,\mathrm{gcl}}}\frac{\partial C_{\mathrm{gcl}}\left(z,t\right)}{\partial z}-\lambda_{\mathrm{gcl}}C_{\mathrm{gcl}}\left(z,t\right)$ $\frac{\partial C_{al}\left(z,t\right)}{\partial t}=\frac{D_{al}}{R_{d,al}}\frac{\partial^{2}C_{al}\left(z,t\right)}{\partial z^{2}}-\frac{v_{al}}{R_{d,al}}\frac{\partial C_{al}\left(z,t\right)}{\partial z}-\lambda C_{al}\left(z,t\right)$

Notes: *C_gm_* and *C_gcl_* (g/L) are the pollutant concentrations in GM and GCL, respectively; *C_gm_(z,t)*, *C_gcl_(z,t)*, and *C_al_(z,t)* (g/L) are the pollutant concentrations in GM, GCL, and AL at time *t* and coordinate *z*, respectively. ① Diffusion: *D_gm_*, *D_gcl_*, and *D_al_* (m^2^/s) are the effective diffusion coefficients of the pollutants in GM, GCL, and AL, respectively; *R_d,gcl_*, *R_d,al_* are the blocking factors of AL and GCL, respectively, and their expressions are $R_{d,\mathrm{gcl}}=1+\frac{\rho_{d,\mathrm{gcl}}K_{d,\mathrm{gcl}}}{n_{\mathrm{gcl}}}$ and $R_{d,\mathrm{al}}=1+\frac{\rho_{d,\mathrm{al}}K_{d,\mathrm{al}}}{n_{\mathrm{al}}}$, where ρd, n, and Kd represent the dry density, porosity, and distribution coefficient of each layer, respectively; *K_g_* is the partition coefficient of the GM and leachate interface. ② Convection: *v_a_* (m/s) is the Darcy flow velocity of convective leakage through holes, and the expressions are $v_{a}=m_{h}Q/A$, $Q=\frac{2h_{w}L_{w}}{l}\left(kb+\sqrt{kl\theta }\right)$, $k=\frac{L_{\mathrm{gcl}}+L_{\mathrm{al}}}{L_{\mathrm{gcl}}/k_{\mathrm{gcl}}+L_{\mathrm{al}}/k_{\mathrm{al}}}$, where *m_h_* is the number of holes on GM, *A* (m^2^) is the area of liner, *Q* is the leakage rate through the hole connected with a wrinkle, *L_w_* (m) is the length of connecting wrinkle, *2b* (m) is the width of connecting wrinkle, *h_w_* (m) is the head difference of leachate, *l* (m) is the sum of the thickness of GCL and AL, *K* (m^2^/s) is the harmonic average permeability coefficient of GCL and AL, *θ* (m^2^/s) is the hydraulic conductivity between GM and GCL, and *v_gcl_* and *v_al_* (m/s) are the seepage velocities in GCL and AL, respectively. ③ Degradation: *λ* is the degradation rate constant, and the expressions are $\lambda_{\mathrm{gcl}}=\ln2/t_{\mathrm{gcl},1/2}$, $\lambda_{\mathrm{al}}=\ln2/t_{\mathrm{al},1/2}$, where *t_gcl,1/2_* and *t_al,1/2_* represent the half-life times of organic pollutants in GCL and AL, respectively.

**Table 6 membranes-12-00922-t006:** Expressions of breakthrough time of pollutants through composite liners.

Various Studies	Expressions of Breakthrough Time
① Chen et al. [19] deduced the breakthrough time expression of GM/GCL/AL considering the effects of convection and diffusion.	$t_{b}=\frac{T_{R}\;R_{d,al\;}\left(z-L_{gcl}-L_{gm}\right)}{v_{al}}$
	where *T_R_* is the time factor, *R_d,al_* is the retardation factor of AL, *L_gcl_* (m) is the thickness of GCL, *L_gm_* (m) is the thickness of GM, and *v_al_* (m/s) is the seepage velocity of the AL layer.
② Xie et al. [111] deduced the breakthrough time expression of GM/GCL/AL considering the effects of convection, diffusion, and degradation.	$t_{b}=\frac{T_{R}R_{d,al}{\left(z-L_{gcl}-L_{gmb}\right)}^{2}}{D_{al}}$
	Other parameters are the same as in the ② expression; *D_al_* (m2/s) is the effective diffusion coefficient of pollutants in AL.
③ Shu et al. [20] deduced the breakthrough time expression of GM/CCL composite liner with COD as the indicator pollutant index.	$t_{b}=\frac{nR_{dc}{\left(l_{g}\sqrt{m}+l_{c}\right)}^{2}}{\left(h+l_{g}\sqrt{m}+l_{c}\right)k_{s}}$ $\left(a{\left(\frac{C_{0}}{C_{A}}\right)}^{b}{\left(\frac{\left(h+l_{g}\sqrt{m}+l_{c}\right)k_{s}}{D_{hc}n}\right)}^{c\ln\left(\frac{C_{0}}{C_{A}}\right)+d}\right)$
	where *n* is the porosity of CCL, R_dc_ is the retardation factor of CCL, *l_g_* and *l_c_* (m) are the GM and CCL thickness, respectively; *m* is defined as *D_Ac_/D_Ag_*, which is the ratio of the diffusion coefficients of CCL and GM; *h* (m) is the head of leachate; *k_s_* (m^2^/s) is the hydraulic conductivity of the equivalent composite liner; *D_hc_* (m^2^/s) is hydrodynamic dispersion coefficient; *C_0_/C_A_* is the relative concentration ratio; *a*, *b*, *c*, and *d* are the state parameters with values of 0.3781, −0.2968, 0.05135, and 0.3305, respectively.
④ Lin et al. [113] deduced the general expression of anti-pollution barrier breakthrough time according to the thermal penetration theory.	$t_{b}=\frac{xv+2u^{2}D-\sqrt{{\left(xv+2u^{2}D\right)}^{2}-v^{2}x^{2}}}{v^{2}}$
	where *x* (m) is the distance from the measuring point to the pollution source, *v* (m/s) is the seepage velocity of groundwater (m/d), *u* is a function of *(x, t)*, and *D* (m^2^/s) is the hydrodynamic dispersion coefficient of pollutants in porous media.

## Data Availability

The data presented in this study are available within this article.

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
