# Peer review of "Review of the Anti-Pollution Performance of Triple-Layer GM/GCL/AL Composite Liners"

_membranes, 2022, doi:10.3390/membranes12100922_

Round 1
Reviewer 1 Report
Please find the attachment.

Reviewer 2 Report
The authors have nicely structured the work and processed certain aspects for the application of three-layer liners. I suggested to the editors that the work be published, but there is only one comment.
When equations are mentioned and values are given, it is necessary to put the units of measurement, for example, whether the equation includes mass in kilograms or grams, etc. So under each equation, when describing individual values, write the units of measurement, for example, m (kg).
Reviewer 3 Report
The manuscript by Li et al. introduced the evaluation and anti-pollution performance of GM/GCL/AL composite liners that are used in landfills to prevent leachate intrusion/contamination towards groundwater. A lot of information was provided by summarizing previous literature, but necessary opinions/views/discussion are required for a review paper. Some general comments to this manuscript are given below:
1. Overall, the whole manuscript requires careful attention to the English grammar/expression. Just as an example, “underground water” in Abstract should be “groundwater”. Lines 73-74 are difficult to understand.
2. The manuscript should update references for the last five years to offer research insights and trends regarding the liners, especially on the aspects of pollutant convection, diffusion, and adsorption.
3. In general, for Sections 2-6, too much information is listed. A better summary is needed. Also, the necessary views and discussions are lacked. Just as an example, the authors summarized many adsorption models in Table 3. What are the major differences? How about the advantages of one over others, so readers can choose based on certain parameters/conditions?
4. The figures were obtained directly from previous papers, which is not suitable for a review paper. Authors can either make their own figures with adaptations from papers or just list references.
Round 2
Reviewer 1 Report
The efforts of the authors are appreciated. They have carefully applied everything that was asked.
Having said that, in my opinion the article is ready for publication in the valuable journal of "membrane".
Author Response
We acknowledge your previous comments and constructive suggestions very much. Thank you for your recognition of our paper now.
Reviewer 3 Report
The authors added some explanations and views to sections 2-7, but more deep insights are expected. Currently, authors typically replace the old reference by recent ones without much deep insights. Those insights and research trends are expected to be provided.
The concern about figure citations remains. The direct copy of figures from previous papers needs permission from authors/editors due to copyright.
Round 3
Reviewer 3 Report
The added analysis and explanation to sections 2-7 seem reasonable and improve the quality of this manuscript.